# Cross-Domain Matching for Bag-of-Words Data via Kernel Embeddings of Latent Distributions

**Yuya Yoshikawa**[*]
Nara Institute of Science and Technology
Nara, 630-0192, Japan
yoshikawa.yuya.yl9@is.naist.jp

**Tomoharu Iwata**
NTT Communication Science Laboratories
Kyoto, 619-0237, Japan
iwata.tomoharu@lab.ntt.co.jp

**Hiroshi Sawada**
NTT Service Evolution Laboratories
Kanagawa, 239-0847, Japan
sawada.hiroshi@lab.ntt.co.jp

**Takeshi Yamada**
NTT Communication Science Laboratories
Kyoto, 619-0237, Japan
yamada.tak@lab.ntt.co.jp

## Abstract

We propose a kernel-based method for finding matching between instances across different domains, such as multilingual documents and images with annotations. Each instance is assumed to be represented as a multiset of features, e.g., a bag-of-words representation for documents. The major difficulty in finding cross-domain relationships is that the similarity between instances in different domains cannot be directly measured. To overcome this difficulty, the proposed method embeds all the features of different domains in a shared latent space, and regards each instance as a distribution of its own features in the shared latent space. To represent the distributions efficiently and nonparametrically, we employ the framework of the kernel embeddings of distributions. The embedding is estimated so as to minimize the difference between distributions of paired instances while keeping unpaired instances apart. In our experiments, we show that the proposed method can achieve high performance on finding correspondence between multi-lingual Wikipedia articles, between documents and tags, and between images and tags.

## 1 Introduction

The discovery of matched instances in different domains is an important task, which appears in natural language processing, information retrieval and data mining tasks such as finding the alignment of cross-lingual sentences [1], attaching tags to images [2] or text documents [3], and matching user identifications in different databases [4].

When given an instance in a source domain, our goal is to find the instance in a target domain that is the most closely related to the given instance. In this paper, we focus on a supervised setting, where correspondence information between some instances in different domains is given. To find matching in a single domain, e.g., find documents relevant to an input document, a similarity (or distance) measure between instances can be used. On the other hand, when trying to find matching between instances in different domains, we cannot directly measure the distances since they consist of different types of features. For example, when matching documents in different languages, since the documents have different vocabularies we cannot directly measure the similarities between documents across different languages without dictionaries.

---

[*]The author moved to Software Technology and Artificial Intelligence Research Laboratory (STAIR Lab) at Chiba Institute of Technology, Japan.

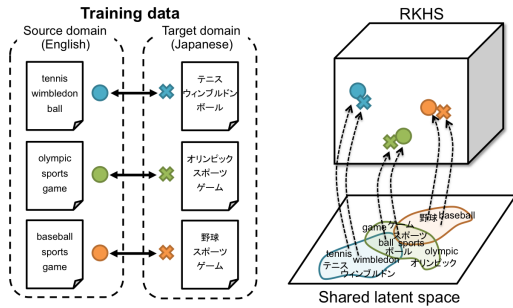

Figure 1: An example of the proposed method used on a multilingual document matching task. Correspondences between instances in source (English) and target (Japanese) domains are observed. The proposed method assumes that each feature (vocabulary term) has a latent vector in a shared latent space, and each instance is represented as a distribution of the latent vectors of the features associated with the instance. Then, the distribution is mapped as an element in a reproducing kernel Hilbert space (RKHS) based on the kernel embeddings of distributions. The latent vectors are estimated so that the paired instances are embedded closer together in the RKHS.

One solution is to map instances in both the source and target domains into a shared latent space. One such method is canonical correspondence analysis (CCA) [5], which maps instances into a latent space by linear projection to maximize the correlation between paired instances in the latent space. However, in practice, CCA cannot solve non-linear relationship problems due to its linearity. To find non-linear correspondence, kernel CCA [6] can be used. It has been reported that kernel CCA performs well as regards document/sentence alignment between different languages [7, 8], when searching for images from text queries [9] and when matching 2D-3D face images [10]. Note that the performance of kernel CCA depends on how appropriately we define the kernel function for measuring the similarity between instances within a domain. Many kernels, such as linear, polynomial and Gaussian kernels, cannot consider the occurrence of different but semantically similar words in two instances because these kernels use the inner-product between the feature vectors representing the instances. For example, words, 'PC' and 'Computer', are different but indicate the same meaning. Nevertheless, the kernel value between instances consisting only of 'PC' and consisting only of 'Computer' is equal to zero with linear and polynomial kernels. Even if a Gaussian kernel is used, the kernel value is determined only by the vector length of the instances.

In this paper, we propose a kernel-based cross-domain matching method that can overcome the problem of kernel CCA. Figure 1 shows an example of the proposed method. The proposed method assumes that each feature in source and target domains is associated with a latent vector in a shared latent space. Since all the features are mapped into the latent space, the proposed method can measure the similarity between features in different domains. Then, each instance is represented as a distribution of the latent vectors of features that are contained in the instance. To represent the distributions efficiently and nonparametrically, we employ the framework of the kernel embeddings of distributions, which measures the difference between distributions in a reproducing kernel Hilbert space (RKHS) without the need to define parametric distributions. The latent vectors are estimated by minimizing the differences between the distributions of paired instances while keeping unpaired instances apart. The proposed method can discover unseen matching in test data by using the distributions of the estimated latent vectors. We will explain matching between two domains below, however, the proposed method can be straightforwardly extended to matching between three and more domains by regarding one of the domains as a pivot domain.

In our experiments, we demonstrate the effectiveness of our proposed method in tasks that involve finding the correspondence between multi-lingual Wikipedia articles, between documents and tags, and between images and tags, by comparison with existing linear and non-linear matching methods.

## 2 Related Work

As described above, canonical correlation analysis (CCA) and kernel CCA have been successfully used for finding various types of cross-domain matching. When we want to match cross-domain instances represented by bag-of-words such as documents, bilingual topic models [1, 11] can also be used. The difference between the proposed method and these methods is that since the proposed method represents each instance as a set of latent vectors of its own features, the proposed method can learn a more complex representation of the instance than these existing methods that represent

each instance as a single latent vector. Another difference is that the proposed method employs a *discriminative* approach, while kernel CCA and bilingual topic models employ *generative* ones.

To model cross-domain data, deep learning and neural network approaches have been recently proposed [12, 13]. Unlike such approaches, the proposed method performs non-linear matching without deciding the number of layers of the networks, which largely affects their performances.

A key technique of the proposed method is the kernel embeddings of distributions [14], which can represent a distribution as an element in an RKHS, while preserving the moment information of the distribution such as the mean, covariance and higher-order moments without density estimation. The kernel embeddings of distributions have been successfully used for a statistical test of the independence of two sample sets [15], discriminative learning on distribution data [16], anomaly detection for group data [17], density estimation [18] and a three variable interaction test [19]. Most previous studies about the kernel embeddings of distributions consider cases where the distributions are unobserved but the samples generated from the distributions are observed. Additionally, each of the samples is represented as a dense vector. With the proposed method, the kernel embedding technique cannot be used to represent the observed multisets of features such as bag-of-words for documents, since each of the features is represented as a one-hot vector whose dimensions are zero except for the dimension indicating that the feature has one. In this study, we benefit from the kernel embeddings of distributions by representing each feature as a dense vector in a shared latent space. The proposed method is inspired by the use of the kernel embeddings of distributions in bag-of-words data classification [20] and regression [21]. Their methods can be applied to single domain data, and the latent vectors of features are used to measure the similarity between the features in a domain. Unlike these methods, the proposed method is used for the cross-domain matching of two different types of domain data, and the latent vectors are used for measuring the similarity between the features in different domains.

## 3 Kernel Embeddings of Distributions

In this section, we introduce the framework of the kernel embeddings of distributions. The kernel embeddings of distributions are used to embed any probability distribution $\mathbb{P}$ on space $\mathcal{X}$ into a reproducing kernel Hilbert space (RKHS) $\mathcal{H}_k$ specified by kernel $k$, and the distribution is represented as element $m^*(\mathbb{P})$ in the RKHS. More precisely, when given distribution $\mathbb{P}$, the kernel embedding of the distribution $m^*(\mathbb{P})$ is defined as follows:

$$m^*(\mathbb{P}) := \mathbb{E}_{\mathbf{x} \sim \mathbb{P}}[k(\cdot, \mathbf{x})] = \int_{\mathcal{X}} k(\cdot, \mathbf{x}) d\mathbb{P} \in \mathcal{H}_k, \tag{1}$$

where kernel $k$ is referred to as *embedding kernel*. It is known that kernel embedding $m^*(\mathbb{P})$ preserves the properties of probability distribution $\mathbb{P}$ such as the mean, covariance and higher-order moments by using *characteristic kernels* (e.g., Gaussian RBF kernel) [22].

When a set of samples $\mathbf{X} = \{\mathbf{x}_l\}_{l=1}^n$ is drawn from the distribution, by interpreting sample set $\mathbf{X}$ as empirical distribution $\hat{\mathbb{P}} = \frac{1}{n} \sum_{l=1}^n \delta_{\mathbf{x}_l}(\cdot)$, where $\delta_{\mathbf{x}}(\cdot)$ is the Dirac delta function at point $\mathbf{x} \in \mathcal{X}$, empirical kernel embedding $m(\mathbf{X})$ is given by

$$m(\mathbf{X}) = \frac{1}{n} \sum_{l=1}^n k(\cdot, \mathbf{x}_l), \tag{2}$$

which can be approximated with an error rate of $||m(\mathbf{X}) - m^*(\mathbb{P})||_{\mathcal{H}_k} = O_p(n^{-\frac{1}{2}})$ [14]. Unlike kernel density estimation, the error rate of the kernel embeddings is independent of the dimensionality of the given distribution.

### 3.1 Measuring Difference between Distributions

By using the kernel embedding representation Eq. (2), we can measure the difference between two distributions. Given two sets of samples $\mathbf{X} = \{\mathbf{x}_l\}_{l=1}^n$ and $\mathbf{Y} = \{\mathbf{y}_{l'}\}_{l'=1}^{n'}$ where $\mathbf{x}_l$ and $\mathbf{y}_{l'}$ belong to the same space, we can obtain their kernel embedding representations $m(\mathbf{X})$ and $m(\mathbf{Y})$. Then, the difference between $m(\mathbf{X})$ and $m(\mathbf{Y})$ is given by

$$D(\mathbf{X}, \mathbf{Y}) = ||m(\mathbf{X}) - m(\mathbf{Y})||_{\mathcal{H}_k}^2. \tag{3}$$

Intuitively, it reflects the difference in the moment information of the distributions. The difference is equivalent to the square of *maximum mean discrepancy (MMD)*, which is used for a statistical test

of independence of two distributions [15]. The difference can be calculated by expanding Eq. (3) as follows:

$$||m(\mathbf{X}) - m(\mathbf{Y})||_{\mathcal{H}_k}^2 = \langle m(\mathbf{X}), m(\mathbf{X}) \rangle_{\mathcal{H}_k} + \langle m(\mathbf{Y}), m(\mathbf{Y}) \rangle_{\mathcal{H}_k} - 2 \langle m(\mathbf{X}), m(\mathbf{Y}) \rangle_{\mathcal{H}_k}, \quad (4)$$

where, $\langle \cdot, \cdot \rangle_{\mathcal{H}_k}$ is an inner-product in the RKHS. In particular, $\langle m(\mathbf{X}), m(\mathbf{Y}) \rangle_{\mathcal{H}_k}$ is given by

$$\langle m(\mathbf{X}), m(\mathbf{Y}) \rangle_{\mathcal{H}_k} = \left\langle \frac{1}{n} \sum_{l=1}^{n} k(\cdot, \mathbf{x}_l), \frac{1}{n'} \sum_{l'=1}^{n'} k(\cdot, \mathbf{y}_{l'}) \right\rangle_{\mathcal{H}_k} = \frac{1}{nn'} \sum_{l=1}^{n} \sum_{l'=1}^{n'} k(\mathbf{x}_l, \mathbf{y}_{l'}). \quad (5)$$

$\langle m(\mathbf{X}), m(\mathbf{X}) \rangle_{\mathcal{H}_k}$ and $\langle m(\mathbf{Y}), m(\mathbf{Y}) \rangle_{\mathcal{H}_k}$ can also be calculated by Eq. (5).

## 4 Proposed Method

Suppose that we are given a training set consisting of $N$ instance pairs $\mathcal{O} = \{(d_i^s, d_i^t)\}_{i=1}^N$, where $d_i^s$ is the $i$th instance in a source domain and $d_i^t$ is the $i$th instance in a target domain. These instances $d_i^s$ and $d_i^t$ are represented as multisets of features included in source feature set $\mathcal{F}^s$ and target feature set $\mathcal{F}^t$, respectively. This means that these instances are represented as bag-of-words (BoW). The goal of our task is to determine the unseen relationship between instances across source and target domains in test data. The number of instances in the source domain may be different to that in the target domain.

### 4.1 Kernel Embeddings of Distributions in a Shared Latent Space

As described in Section 1, the difficulty as regards finding cross-domain instance matching is that the similarity between instances across source and target domains cannot be directly measured. We have also stated that although we can find a latent space that can measure the similarity by using kernel CCA, standard kernel functions, e.g., a Gaussian kernel, cannot reflect the co-occurrence of different but related features in a kernel calculation between instances. To overcome them, we propose a new data representation for finding cross-domain instance matching. The proposed method assumes that each feature in a source feature set, $f \in \mathcal{F}^s$, has a $q$-dimensional latent vector $\mathbf{x}_f \in \mathbb{R}^q$ in a shared space. Likewise, each feature in target feature set, $g \in \mathcal{F}^t$, has a $q$-dimensional latent vector $\mathbf{y}_g \in \mathbb{R}^q$ in the shared space. Since all the features in the source and target domains are mapped into a common shared space, the proposed method can capture the relationship between features both in each domain and across different domains. We define the sets of latent vectors in the source and target domains as $\mathbf{X} = \{\mathbf{x}_f\}_{f \in \mathcal{F}^s}$ and $\mathbf{Y} = \{\mathbf{y}_g\}_{g \in \mathcal{F}^t}$, respectively.

The proposed method assumes that each instance is represented by a distribution (or multiset) of the latent vectors of the features that are contained in the instance. The $i$th instance in the source domain $d_i^s$ is represented by a set of latent vectors $\mathbf{X}_i = \{\mathbf{x}_f\}_{f \in d_i^s}$ and the $j$th instance in the target domain $d_j^t$ is represented by a set of latent vectors $\mathbf{Y}_j = \{\mathbf{y}_g\}_{g \in d_j^t}$. Note that $\mathbf{X}_i$ and $\mathbf{Y}_j$ lie in the same latent space.

In Section 3, we introduced the kernel embedding representation of a distribution and described how to measure the difference between two distributions when samples generated from the distribution are observed. In the proposed method, we employ the kernel embeddings of distributions to represent the distributions of the latent vectors for the instances. The kernel embedding representations for the $i$th source and the $j$th target domain instances are given by

$$m(\mathbf{X}_i) = \frac{1}{|d_i^s|} \sum_{f \in d_i^s} k(\cdot, \mathbf{x}_f), \qquad m(\mathbf{Y}_j) = \frac{1}{|d_j^t|} \sum_{g \in d_j^t} k(\cdot, \mathbf{y}_g). \quad (6)$$

Then, the difference between the distributions of the latent vectors are measured by using Eq. (3), that is, the difference between the $i$th source and the $j$th target domain instances is given by

$$D(\mathbf{X}_i, \mathbf{Y}_j) = ||m(\mathbf{X}_i) - m(\mathbf{Y}_j)||_{\mathcal{H}_k}^2. \quad (7)$$

### 4.2 Model

The proposed method assumes that paired instances have similar distributions of latent vectors and unpaired instances have different distributions. In accordance with the assumption, we define the likelihood of the relationship between the $i$th source domain instance and the $j$th target domain instance as follows:

$$p(d_j^t | d_i^s, \mathbf{X}, \mathbf{Y}, \theta) = \frac{\exp\left(-D(\mathbf{X}_i, \mathbf{Y}_j)\right)}{\sum_{j'=1}^{N} \exp\left(-D(\mathbf{X}_i, \mathbf{Y}_{j'})\right)}, \quad (8)$$

where, $\theta$ is a set of hyper-parameters for the embedding kernel used in Eq. (6). Eq. (8) is in fact the conditional probability with which the $j$th target domain instance is chosen given the $i$th source domain instance. This formulation is more efficient than we consider a bidirectional matching. Intuitively, when distribution $\mathbf{X}_i$ is more similar to $\mathbf{Y}_j$ than other distributions $\{\mathbf{Y}_{j'} \mid j' \neq j\}_{j'=1}^N$, the probability has a higher value.

We define the posterior distribution of latent vectors $\mathbf{X}$ and $\mathbf{Y}$. By placing Gaussian priors with precision parameter $\rho > 0$ for $\mathbf{X}$ and $\mathbf{Y}$, that is, $p(\mathbf{X}|\rho) \propto \prod_{\mathbf{x} \in \mathbf{X}} \exp\left(-\frac{\rho}{2}||\mathbf{x}||_2^2\right), p(\mathbf{Y}|\rho) \propto \prod_{\mathbf{y} \in \mathbf{Y}} \exp\left(-\frac{\rho}{2}||\mathbf{y}||_2^2\right)$, the posterior distribution is given by

$$p(\mathbf{X}, \mathbf{Y}|\mathcal{O}, \Theta) = \frac{1}{Z} p(\mathbf{X}|\rho) p(\mathbf{Y}|\rho) \prod_{i=1}^N p(d_i^t|d_i^s, \mathbf{X}, \mathbf{Y}, \theta), \tag{9}$$

where, $\mathcal{O} = \{(d_i^s, d_i^t)\}_{i=1}^N$ is a training set of $N$ instance pairs, $\Theta = \{\theta, \rho\}$ is a set of hyper-parameters and $Z = \int\int p(\mathbf{X}, \mathbf{Y}, \mathcal{O}, \Theta) d\mathbf{X} d\mathbf{Y}$ is a marginal probability, which is constant with respect to $\mathbf{X}$ and $\mathbf{Y}$.

### 4.3 Learning

We estimate latent vectors $\mathbf{X}$ and $\mathbf{Y}$ by maximizing the posterior probability of the latent vectors given by Eq. (9). Instead of Eq. (9), we consider the following negative logarithm of the posterior probability,

$$\mathcal{L}(\mathbf{X}, \mathbf{Y}) = \sum_{i=1}^N \left\{ D(\mathbf{X}_i, \mathbf{Y}_i) + \log \sum_{j=1}^N \exp\left(-D(\mathbf{X}_i, \mathbf{Y}_j)\right) \right\} + \frac{\rho}{2} \left( \sum_{\mathbf{x} \in \mathbf{X}} ||\mathbf{x}||_2^2 + \sum_{\mathbf{y} \in \mathbf{Y}} ||\mathbf{y}||_2^2 \right), \tag{10}$$

and minimize it with respect to the latent vectors. Here, maximizing Eq. (9) is equivalent to minimizing Eq. (10). To minimize Eq. (10) with respect to $\mathbf{X}$ and $\mathbf{Y}$, we perform a gradient-based optimization. The gradient of Eq. (10) with respect to each $\mathbf{x}_f \in \mathbf{X}$ is given by

$$\frac{\partial \mathcal{L}(\mathbf{X}, \mathbf{Y})}{\partial \mathbf{x}_f} = \sum_{i:f \in d_i^s} \left\{ \frac{\partial D(\mathbf{X}_i, \mathbf{Y}_i)}{\partial \mathbf{x}_f} - \frac{1}{c_i} \sum_{j=1}^N e_{ij} \frac{\partial D(\mathbf{X}_i, \mathbf{Y}_j)}{\partial \mathbf{x}_f} \right\} + \rho \mathbf{x}_f \tag{11}$$

where,

$$e_{ij} = \exp\left(-D(\mathbf{X}_i, \mathbf{Y}_j)\right), \qquad c_i = \sum_{j=1}^N \exp\left(-D(\mathbf{X}_i, \mathbf{Y}_j)\right), \tag{12}$$

and the gradient of the difference between distributions $\mathbf{X}_i$ and $\mathbf{Y}_j$ with respect to $\mathbf{x}_f$ is given by

$$\frac{\partial D(\mathbf{X}_i, \mathbf{Y}_j)}{\partial \mathbf{x}_f} = \frac{1}{|d_i^s|^2} \sum_{l \in d_i^s} \sum_{l' \in d_i^s} \frac{\partial k(\mathbf{x}_l, \mathbf{x}_{l'})}{\partial \mathbf{x}_f} - \frac{2}{|d_i^s||d_j^t|} \sum_{l \in d_i^s} \sum_{g \in d_i^t} \frac{\partial k(\mathbf{x}_l, \mathbf{y}_g)}{\partial \mathbf{x}_f}. \tag{13}$$

When the distribution $\mathbf{X}_i$ does not include the latent vector $\mathbf{x}_f$, the gradient consistently becomes a zero vector. $\frac{\partial k(\mathbf{x}_l, \mathbf{x}_{l'})}{\partial \mathbf{x}_f}$ is the gradient of an embedding kernel. This depends on the choice of kernel. When the embedding kernel is a Gaussian kernel, the gradient is calculated as with Eq. (15) in [21]. Similarly, The gradient of Eq. (10) with respect to each $\mathbf{y}_g \in \mathbf{Y}$ is given by

$$\frac{\partial \mathcal{L}(\mathbf{X}, \mathbf{Y})}{\partial \mathbf{y}_g} = \sum_{i=1}^N \left\{ \frac{\partial D(\mathbf{X}_i, \mathbf{Y}_i)}{\partial \mathbf{y}_g} - \frac{1}{c_i} \sum_{j:g \in d_j^t} e_{ij} \frac{\partial D(\mathbf{X}_i, \mathbf{Y}_j)}{\partial \mathbf{y}_g} \right\} + \rho \mathbf{y}_g, \tag{14}$$

where, the gradient of the difference between distributions $\mathbf{X}_i$ and $\mathbf{Y}_j$ with respect to $\mathbf{y}_g$ can be calculated as with Eq. (13)

Learning is performed by alternately updating $\mathbf{X}$ using Eq. (11) and updating $\mathbf{Y}$ using Eq. (14) until the improvement in the negative log likelihood Eq. (10) converges.

### 4.4 Matching

After the estimation of the latent vectors $\mathbf{X}$ and $\mathbf{Y}$, the proposed method can reveal the matching between test instances. The matching is found by first measuring the difference between a given source domain instance and target domain instances using Eq. (7), and then searching for the instance pair with the smallest difference.

# 5 Experiments

In this section, we report our experimental results for three different types of cross-domain datasets: multi-lingual Wikipedia, document-tag and image-tag datasets.

**Setup of proposed method.** Throughout these experiments, we used a Gaussian kernel with parameter $\gamma \geq 0$: $k(\mathbf{x}_f, \mathbf{y}_g) = \exp\left(-\frac{\gamma}{2}||\mathbf{x}_f - \mathbf{y}_g||_2^2\right)$ as an embedding kernel. The hyper-parameters of the proposed method are the dimensionality of a shared latent space $q$, a regularizer parameter for latent vectors $\rho$ and a Gaussian embedding kernel parameter $\gamma$. After we train the proposed method with various hyper-parameters $q \in \{8, 10, 12\}$, $\rho \in \{0, 10^{-2}, 10^{-1}\}$ and $\gamma \in \{10^{-1}, 10^0, \cdots, 10^3\}$, we chose the optimal hyper-parameters by using validation data. When training the proposed method, we initialized latent vectors $\mathbf{X}$ and $\mathbf{Y}$ by applying principle component analysis (PCA) to a matrix concatenating two feature-frequency matrices in the source and target domains. Then, we employed the L-BFGS method [23] with gradients given by Eqs. (11) (14) to learn the latent vectors.

**Comparison methods.** We compared the proposed method with the $k$-nearest neighbor method (KNN), canonical correspondence analysis (CCA), kernel CCA (KCCA), bilingual latent Dirichlet allocation (BLDA), and kernel CCA with the kernel embeddings of distributions (KED-KCCA). For a test instance in the source domain, our KNN searches for the nearest neighbor source instances in the training data, and outputs a target instance in the test data, which is located close to the target instances that are paired with the searched for source instances. CCA and KCCA first learn the projection of instances into a shared latent space using training data, and then they find matching between instances by projecting the test instances into the shared latent space. KCCA used a Gaussian kernel for measuring the similarity between instances and chose the optimal Gaussian kernel parameter and regularizer parameter by using validation data. With BLDA, we first learned the same model as [1, 11] and found matching between instances in the test data by obtaining the topic distributions of these instances from the learned model. KED-KCCA uses the kernel embeddings of distributions described in Section 3 for obtaining the kernel values between the instances. The vector representations of features were obtained by applying singular value decomposition (SVD) for instance-feature frequency matrices. Here, we set the dimensionality of the vector representations to 100. Then, KED-KCCA learns kernel CCA with the kernel values as with the above KCCA. With CCA, KCCA, BLDA and KED-KCCA, we chose the optimal latent dimensionality (or number of topics) within $\{10, 20, \cdots, 100\}$ by using validation data.

**Evaluation method.** Throughout the experiments, we quantitatively evaluated the matching performance by using the precision with which the true target instance is included in a set of $R$ candidate instances, $\mathcal{S}(R)$, found by each method. More formally, the precision is given by

$$\text{Precision}@R = \frac{1}{N_{\text{te}}} \sum_{i=1}^{N_{\text{te}}} \delta\left(t_i \in \mathcal{S}_i(R)\right), \tag{15}$$

where, $N_{\text{te}}$ is the number of test instances in the target domain, $t_i$ is the $i$th true target instance, $\mathcal{S}_i(R)$ is $R$ candidate instances of the $i$th source instance and $\delta(\cdot)$ is the binary function that returns 1 if the argument is true, and 0 otherwise.

## 5.1 Matching between Bilingual Documents

With a multi-lingual Wikipedia document dataset, we examine whether the proposed method can find the correct matching between documents written in different languages. The dataset includes 34,024 Wikipedia documents for each of six languages: German (de), English (en), Finnish (fi), French (fr), Italian (it) and Japanese (ja), and documents with the same content are aligned across the languages. From the dataset, we create $_6C_2 = 15$ bilingual document pairs. We regard the first component of the pair as a source domain and the other as a target domain. For each of the bilingual document pairs, we randomly create 10 evaluation sets that consist of 1,000 document pairs as training data, 100 document pairs as validation data and 100 document pairs as test data. Here, each document is represented as a bag-of-words without stopwords and low frequency words.

Figure 2 shows the matching precision for each of the bilingual pairs of the Wikipedia dataset. With all the bilingual pairs, the proposed method achieves significantly higher precision than the other methods with a wide range of $R$. Table 1 shows examples of predicted matching with the Japanese-English Wikipedia dataset. Compared with KCCA, which is the second best method, the

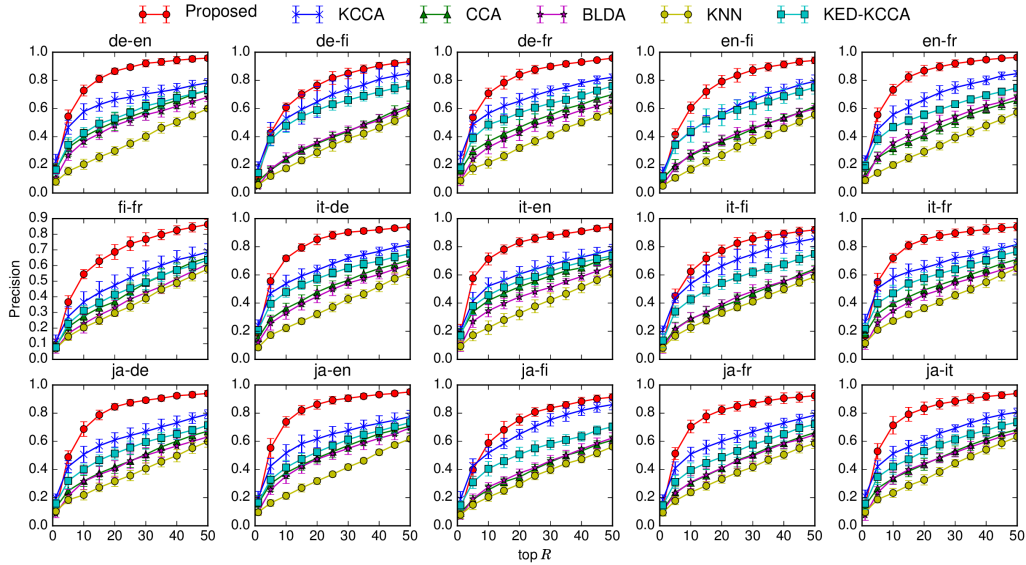

Figure 2: Precision of matching prediction and its standard deviation on multi-lingual Wikipedia datasets.

Table 1: Top five English documents matched by the proposed method and KCCA given five Japanese documents in the Wikipedia dataset. Titles in bold typeface indicate correct matching.

(a) Japanese Input title: SD （SD card）

| Proposed | Intel, **SD card**, Libavcodec, MPlayer, Freeware |
|---|---|
| KCCA | BBC World News, **SD card**, Morocco, Phoenix, 24 Hours of Le Mans |

(b) Japanese Input title: （Anthrax）

| Proposed | Psittacosis, **Anthrax**, Dehydration, Isopoda, Cataract |
|---|---|
| KCCA | Dehydration, Psittacosis, Cataract, Hypergeometric distribution, Long Island Iced Tea |

(c) Japanese Input title: （Doppler effect）

| Proposed | LU deconmposition, Redshift, **Doppler effect**, Phenylalanine, Dehydration |
|---|---|
| KCCA | Long Island Iced Tea, Opportunity cost, Cataract, Hypergeometric distribution, Intel |

(d) Japanese Input title: （Mexican cuisine）

| Proposed | **Mexican cuisine**, Long Island Iced Tea, Phoenix, Baldr, China Radio International |
|---|---|
| KCCA | Taoism, Chariot, Anthrax, Digital Millennium Copyright Act, Alexis de Tocqueville |

(e) Japanese Input title: （Freeware）

| Proposed | BBC World News, Opportunity cost, **Freeware**, NFS, Intel |
|---|---|
| KCCA | Digital Millennium Copyright Act, China Radio International, Hypergeometric distribution, Taoism, Chariot |

proposed method can find both the correct document and many related documents. For example, in Table 1(a), the correct document title is "SD card". The proposed method outputs the SD card's document and documents related to computer technology such as "Intel" and "MPlayer". This is because the proposed method can capture the relationship between words and reflect the difference between documents across different domains by learning the latent vectors of the words.

## 5.2 Matching between Documents and Tags, and between Images and Tags

We performed experiments matching documents and tailgates, and matching images and tailgates with the datasets used in [3]. When matching documents and tailgates, we use datasets obtained from two social bookmarking sites, `delicious`[1] and `hatena`[2], and `patent` dataset. The `delicious` and the `hatena` datasets include pairs consisting of a web page and a tag list labeled by users, and the patent dataset includes pairs consisting of a patent description and a tag list representing the category of the patent. Each web page and each patent description are represented

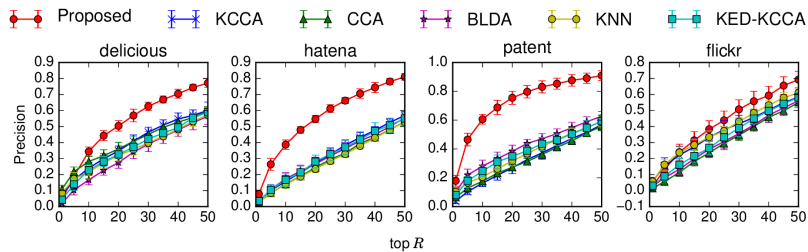

Figure 3: Precision of matching prediction and its standard deviation on `delicious`, `hatena`, `patent` and `flickr` datasets.

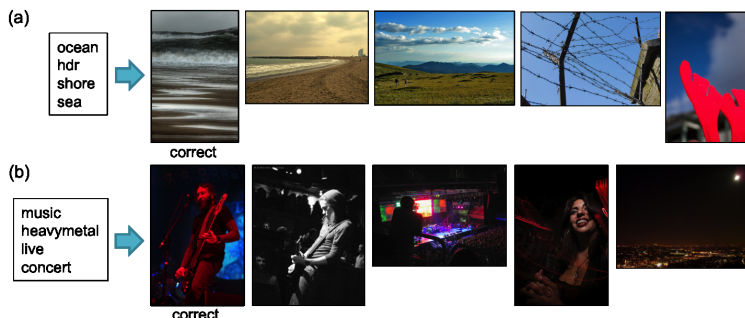

Figure 4: Two examples of input tag lists and the top five images matched by the proposed method on the `flickr` dataset.

as a bag-of-words as with the experiments using the Wikipedia dataset, and the tag list is represented as a set of tags. With the matching of images and tag lists, we use the `flickr` dataset, which consists of pairs of images and tag lists. Each image is represented as a bag-of-visual-words, which is obtained by first extracting features using SIFT, and then applying K-means clustering with 200 components to the SIFT features. For all the datasets, the numbers of training, test and validation pairs are 1,000, 100 and 100, respectively.

Figure 3 shows the precision of the matching prediction of the proposed and comparison methods for the `delicious`, `hatena`, `patent` and `flickr` datasets. The precision of the comparison methods with these datasets was much the same as the precision of random prediction. Nevertheless, the proposed method achieved very high precision particularly for the `delicious`, `hatena` and `patent` datasets. Figure 4 shows examples of input tag lists and the top five images matched by the proposed method with the `flickr` dataset. In the examples, the proposed method found the correct images and similar related images from given tag lists.

## 6    Conclusion

We have proposed a novel kernel-based method for addressing cross-domain instance matching tasks with bag-of-words data. The proposed method represents each feature in all the domains as a latent vector in a shared latent space to capture the relationship between features. Each instance is represented by a distribution of the latent vectors of features associated with the instance, which can be regarded as samples from the unknown distribution corresponding to the instance. To calculate difference between the distributions efficiently and nonparametrically, we employ the framework of kernel embeddings of distributions, and we learn the latent vectors so as to minimize the difference between the distributions of paired instances in a reproducing kernel Hilbert space. Experiments on various types of cross-domain datasets confirmed that the proposed method significantly outperforms the existing methods for cross-domain matching.

**Acknowledgments.** This work was supported by JSPS Grant-in-Aid for JSPS Fellows (259867).

## Footnotes

[1] https://delicious.com/

[2] http://b.hatena.ne.jp/

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
