[Reviews · NeurIPS 2015]

Submitted by Assigned_Reviewer_1

%%%% post-rebuttal comment %%%%

I have increased the score from 6 to 7 to better reflect my impression of this paper.

%%%%

== Summary ==

This paper proposes a kernel-based method for cross-domain matching for bag-of-words data. The difficulty is that the data may be represented using features across domains so that they cannot be compared directly. A common approach is to learn a "shared latent space". Existing algorithms such as CCA, KCCA, and topic modeling can be thought in this way. Unlike previous works, this paper looks at the data, e.g., documents and images, as a probability distribution over vectors in the shared latent space. As a result, the proposed method is able to capture the occurrence of different but semantically similar features in two distinct instances. The latent vectors are estimated by maximizing the the posterior probability of the latent vectors expressed in terms of the RKHS distance between instances (aka maximum mean discrepancy). Experimental results on several real-world datasets, e.g., bilingual documents, document-tag matching, and image-tag matching, are very encouraging.

In summary, the paper proposes a simple and intuitive idea for cross-domain matching problem with strong empirical results.

== Quality ==

The key weakness of this paper is the technicality.

== Clarity ==

The paper is clearly written. I only have minor comments on clarity.

In the experiment section, you mentioned the "development data". Does it refer to the validation data? [Line 185--189] This paragraph seems to suggest that X_i and Y_j live in different latent spaces. It should be clarified. Eq. (9) Why do you consider this prior? What is the motivation? It should be clarified. In Eq. (11), perhaps you could mention that the prior you chose will correspond to the L_2 regularization on x and y. This is also in line with the formulation considered in Yuya et al., (NIPS2014). Again, it should be clarified why this is a good choice.

== Originality ==

The proposed idea seems to be quite similar to Yuya et al., (NIPS2014), but considers the cross-domain matching problem instead of classification. Nevertheless, I think the idea of treating data as probability distributions over latent space is quite interesting and has potential applications in many areas.

== Significance ==

I think the idea presented in this work can be applied more broadly. For example, this could be used for "representation learning" in which the latent space may have a hierarchical structure, e.g., tree, or represent the parameters of some generative model, e.g., RBM or deep neural network.

Relevant papers:

Y. Li, K. Swersky, R. Zemel. Generative Moment Matching Networks. ICML 2015. G.K. Dziugaite, D. Roy, Z. Ghahramani. Training generative neural networks via Maximum Mean Discrepancy optimization. UAI 2015.

Minor comments

Eq. (15) and (16) seem to be redundant as they are basically the same as Eq. (12) and (14). Maybe they can be removed to save space.
Summary: A simple and intuitive approach for cross-domain matching of bag-of-words data with strong empirical results.

Submitted by Assigned_Reviewer_2

Comments after response:

Given that [18] saw only mild absolute improvements over the two-stage approach, I certainly don't think it's clear that any combined approach will so outperform two-stage approaches that you don't need to run the experiment. Especially given that many modern NLP systems will be using word embeddings anyway, I think it's important to see comparisons to natural approaches employing them.

In addition to running CCA on separate word embeddings, you should also compare to matching by simply looking for the nearest candidate word in a existing multilingual word embedding system. Your technique could be viewed as an approach for multilingual word embedding; thus comparing to the existing literature there is also important, in my opinion.

I would certainly recommend for publication a version of this paper with thorough experimental comparisons to these related approaches, if the experimental results were promising; the novelty over [18] and [19] is perhaps not enormous, but publicizing their application to a new domain is sensible. Without seeing those results, however, it's harder for me to be enthusiastic about the paper. I've increased my score from 4 to 5.

-----

The paper proposes a method for cross-domain matching by embedding instances as sets, and maximizing the likelihood of a softmax model based on MMD distances between those sets given limited should-match training pairs.

Here is an alternative algorithm for this problem: perform word2vec-type word embeddings in each domain, then run kernel CCA on that representation. This algorithm is extremely natural and overcomes this paper's complaints about kernel CCA, and yet nothing of its type is mentioned here.

In general, this paper seems to entirely ignore the *enormous* recent literature on embeddings. Here are five extremely relevant papers I found immediately with a relevant search specifically about the NLP domain:

- Bollegala, Maehara, and Kawarabayashi. Unsupervised Cross-Domain Word Representation Learning. ACL 2015. - Shimodaira. A simple coding for cross-domain matching with dimension reduction via spectral graph embedding. arXiv:1412.8380. - Yang and Eisenstein. Unsupervised Domain Adaptation with Feature Embeddings. ICLR 2015 workshop (arXiv:1412.4385). - Al-Rfou, Perozzi, and Skiena. Polyglot: Distributed Word Representations for Multilingual NLP. arXiv:1307.1662. - Nguyen and Grishman. Employing Word Representations and Regularization for Domain Adaptation of Relation Extraction. ACL 2014 short paper.

Additionally, the huge number of deep learning papers last year about automatic caption generation are quite relevant. Some of these techniques rely on more paired training examples than you use here, but given that you still use some such examples, a more thorough evaluation, or even a brief mention, of that distinction is necessary.

This paper's techniques are intriguing (despite being somewhat incremental over the previous similar papers), but without even discussing their relationship to the relevant literature, it is impossible for a reader to judge how the learned embeddings compare to those of actual competing systems, rather than straw-man baselines.

The method also seems to be quite difficult to scale, as with the previous two papers in this series. This is not discussed, though all datasets evaluated are quite small.
Summary: The proposed method is interesting, but its advantages over very natural alternatives (not discussed in the paper) are not clear, and the enormous literature of related work is barely touched upon.

Submitted by Assigned_Reviewer_3

The idea of cross domain matching based on correlation analysis is known in the literature. However, this paper proposes a novel approach to compute a domain specific shared space of latent distributions that makes it novel.

The most striking aspect of the paper is the experimental results. It shows that the proposed method significantly outperforms other methods in the literature. Additionally, the paper is extremely well written, leaving no need for further improvement.

Summary: The paper describes a method to compare Bag of Words(BoW) vectors from different domains via a kernel embedding method. The method proposed here is elegant and the problem itself is extremely interesting.

Submitted by Assigned_Reviewer_4

The authors propose a kernel-based algorithm for cross domain matching task where the goal is to retrieve a matching instance in the target set given an instance in the source set. Each instance of the two different domains is treated as a bag of features (e.g., for words) and is embedded to the common space (RKHS) with kernel embedding for matching purpose. The features of the two domains are learned jointly by maximizing a likelihood which is (intuitively) inversely proportional to the distance (i.e., MMD) of the embedded instances in RKHS. With real data, the proposed method outperforms standard approaches like CCA and kernel CCA on document-document, document-tag, and tag-image matching tasks.

The writing is clear and easy to follow. The experimental results are impressive as the method consistently outperforms all other methods in all the three experiments. In machine learning (excluding information retrieval), I understand that cross domain matching is mainly tackled by CCA or kernel CCA. So the paper provides a starting point for further exploration in this direction.

Regarding the originality, the idea of embedding a bag of hidden features with kernel and learn the features is, however, not new as this was considered in [18] (Latent Support Measure Machines for Bag-of-Words Data Classification, NIPS 2014). The part which is original seems to be the use of such approach for cross-domain matching with a proposed probabilistic model similar to the one in kernel logistic regression. Although the experimental results provide an evidence that the method performs better than KCCA on certain tasks, the paper does not give enough motivation, justification and description of the advantages of the method for better understanding.

My comments/questions in descending order of importance.

1. From lines 70-78, the problem of kernel CCA is not clearly stated. This is an important motivating paragraph in the paper.

2. Presumably the learned latent vectors x_f, y_g play more role in the algorithm than the kernel k. How does the Gaussian kernel k compare to a linear kernel with a large latent dimension q ? I have a feeling that with large q, the learned latent vectors will have enough degree of freedom to perform the task even with linear kernel. This is to say that, if my intuition is correct, a kernel is not needed.

3. What is the motivation for the likelihood in Eq. 8 ? To me, this appears to be an ad-hoc choice. What is wrong with using just ||m(X)-m(Y)||^2 as the loss in Eq. 11 ? Since the log term in Eq. 11 will disappear, this should make optimization simpler and cheaper.

4. How does the method compare qualitatively (not numerically) with kernel CCA ? I suggest that the authors shorten Sec. 4.1 and Sec. 4.3 and explain it.

5. It would be interesting to see the effect of varying q (latent dimension) on the precision and run time. The experiments only focus on one aspect which is the precision.

6. What is the computational cost of the method as compared to kernel CCA ? It seems kernel CCA can be more efficient. The gradient in Eq. 12 must be expensive and the objective is highly non-convex. This should be addressed in the paper.

7. Line 284-286: Are the hyper-parameters chosen by cross validation ? By "development data", is it the same as a validation set ?

8. In the experiments, does KCCA use the same learned featured x_f, y_g from the proposed method ? If not, what is the kernel of KCCA i.e., Gaussian kernel on what ? What is the result if you do so ?

9. In abstract, lines 27-28, "..while keeping unpaired instances apart." How is this criterion implemented in the method ?

10. Lines 100-102, "..can learn a more complex representation..." How ? If possible, can you show this experimentally ?

Minor:

1. Eq. 3: ||m(X)-m(Y)||^2 is not a distance; ||m(X)-m(Y)|| (or MMD) is.

2. Should mention that x_f is a latent vector for word (feature) f in the experiments.

3. In the references, the format is not consistent. Some entries are written with abbreviated author names.

===== after rebuttal ======= I have considered the authors' rebuttal which answered only some of my questions.

Summary: The paper proposes an interesting kernel-based algorithm for cross domain matching with three real experiments. However, the motivation is not clearly stated and qualitative description of the method is insufficient.

Author Feedback
Author rebuttal: We would like to thank the reviewers for their feedback and insightful comments, which we shall address below.

[Assigned_Reviewer_2]
We would like to answer your questions in numerical order.

> 1
Standard kernels used in kernel CCA only consider the correlation of
the same feature (word) in two data (documents) and cannot capture the
relationship between different features such as `PC' and `computer'
which represent much the same object or `baseball' and `soccer' which
are both sports.
Thus, we need to develop a new kernel for cross-domain bag-of-words data.
We will explain the above motivation clearly.

> 2
In our preliminary experiments, we tried a linear embedding kernel in the proposed method.
However, the matching performance was worse than kernel CCA, even with the latent dimensionality increased up to q=30 (q used in this paper is at most 12).
Thus, using a non-linear kernel is essential to improve the matching performance.

> 3
The likelihood is designed such that paired instances have similar distributions in a shared latent space, while the others have dissimilar distributions.
Instead of minimizing only the loss of paired instances leaving out the information of unpaired instances, we employ the likelihood to utilize the whole information of training data.

> 4
A difference between the proposed method and kernel CCA is that the proposed method can overcome the problem of kernel CCA described in the above answer to Q1 by incorporating latent vectors of features.
Another difference is that the proposed method is *discriminative* matching model.
We will explain it clearly in Sec.4.

> 5
On the `delicious' dataset, we tried varying q within a range {2, 4, ..., 30}.
Actually, the best precision is achieved at q=20, and the precision decreases with increasing q > 20 due to over-fitting.
The time complexity of calculating gradient Eq.(12) is linear with respect to q.

> 6
In this paper, we want to show the effectiveness of the proposed method by focusing on improving matching performance.
Efficient learning for the proposed method can be performed by SGD.

> 7
That's right. We will fix it.

> 8
KCCA uses only BoW features.
Although we can use the latent vectors x_f, y_g as inputs to KCCA, the matching performance would be worse than the proposed method because the objectives of KCCA and the proposed method are different.

> 9
Please see the answer to Q3.

> 10
This sentence means that the proposed method includes more parameters than kernel CCA by employing appropriate representation for BoW data.

[Assigned_Reviewer_4]
> [Line 185--189] ...
X_i and Y_j are sets of latent vectors in the same latent space.
We will explain it clearly.

> Eq. (9) Why do you consider this prior? ...
We used a Gaussian prior because latent vectors are assumed to be in a continuous space and sparse vectors are not needed, and the prior is easy to implement.
By placing a Gaussian prior, it is expected that the latent vectors of features that are not useful for matching are likely to be located close to the origin in the latent space.

> Eq. (15) and (16) seem to be redundant ...
We will reorganize the part in the final paper.

[Assigned_Reviewer_6]
> Here is an alternative algorithm ...
As you mentioned, we can consider the two-stage approach.
However, as reported by [18] which is a basis of our method, the approach would not work well because the objectives of word embedding and the matching model are different.
To further confirm the effectiveness of the proposed method, we will add the experiments of the two-stage approach.

> In general, this paper seems to entirely ignore ...
The papers you mentioned here proposed to learn word embeddings in unsupervised ways.
Since the goal of this work is to improve the performance of cross-domain matching, rather than obtaining the word embeddings, our goal is basically different from their goals.
In terms of matching performance, it is expected that the proposed method is better than the approaches with unsupervised word embeddings.

> Additionally, the huge number of deep learning papers ...
Although there are many studies using cross-domain (or multi-modal) data in deep learning (DL), the kernel-based approach is not well studied.
Unlike DL methods, the proposed method does not need to decide the number of layers, which largely affects its performance.
We will discuss such differences and relationships between DL and the proposed method.

> This paper's techniques are intriguing ...
Please see the previous answer.

> The method also seems to be quite difficult to scale ...
As with DL, more efficient learning for the proposed method can be achieved by applying SGD.
Speed-up techniques for general kernel methods such as kernel approximation may also be applied.
In this paper, we focus on demonstrating the effectiveness of our kernel-based approach in terms of matching accuracy.